# Strategies to Develop a Mucosa-Targeting Vaccine against Emerging Infectious Diseases

**DOI:** 10.3390/v14030520

**Published:** 2022-03-03

**Authors:** Fengling Feng, Ziyu Wen, Jiaoshan Chen, Yue Yuan, Congcong Wang, Caijun Sun

**Affiliations:** 1School of Public Health (Shenzhen), Shenzhen Campus of Sun Yat-sen University, Shenzhen 518107, China; fengfling@mail.sysu.edu.cn (F.F.); wenzy3@mail2.sysu.edu.cn (Z.W.); chenjsh59@mail2.sysu.edu.cn (J.C.); yuany263@mail2.sysu.edu.cn (Y.Y.); wangcc5@mail2.sysu.edu.cn (C.W.); 2Key Laboratory of Tropical Disease Control (Sun Yat-sen University), Ministry of Education, Guangzhou 510080, China

**Keywords:** vaccine, mucosal immunity, emerging infectious diseases

## Abstract

Numerous pathogenic microbes, including viruses, bacteria, and fungi, usually infect the host through the mucosal surfaces of the respiratory tract, gastrointestinal tract, and reproductive tract. The mucosa is well known to provide the first line of host defense against pathogen entry by physical, chemical, biological, and immunological barriers, and therefore, mucosa-targeting vaccination is emerging as a promising strategy for conferring superior protection. However, there are still many challenges to be solved to develop an effective mucosal vaccine, such as poor adhesion to the mucosal surface, insufficient uptake to break through the mucus, and the difficulty in avoiding strong degradation through the gastrointestinal tract. Recently, increasing efforts to overcome these issues have been made, and we herein summarize the latest findings on these strategies to develop mucosa-targeting vaccines, including a novel needle-free mucosa-targeting route, the development of mucosa-targeting vectors, the administration of mucosal adjuvants, encapsulating vaccines into nanoparticle formulations, and antigen design to conjugate with mucosa-targeting ligands. Our work will highlight the importance of further developing mucosal vaccine technology to combat the frequent outbreaks of infectious diseases.

## 1. Mucosal Defense against Pathogenic Microbes

The frequent emergence and re-emergence of highly pathogenic microbes have become a severe threat for human public health, including severe acute respiratory syndrome coronavirus (SARS-CoV), SARS-CoV-2, avian influenza, Ebola virus (EBOV), human immunodeficiency virus (HIV), and mycobacterium tuberculosis (TB). Most of these pathogens infect the host mainly through the mucosal route, such as the skin epithelium, respiratory tract, ocular mucosa, gastrointestinal tract, and reproductive tract (Figure 1). The mucosa surface consequently plays a key role as the first defense system against these pathogens’ invasion, which is composed of physical, chemical, biological, and immunological barriers. The physical barrier mainly consists of the close tight junctions of epithelial cells, cilia oscillations, and mucus produced by goblet cells in mucosal tissue; the chemical barrier can include anti-bacterial proteases, various digestive enzymes, lysozymes, special pH environments, and mucopolysaccharides; the microbiomes (gut flora and genital tract flora) and their metabolites constitute a biological barrier; the mucosal immune barrier is mainly formed by secretory IgA (sIgA), cytokines, mucosa-associated lymphoid tissue (MALT), and diffused innate and adaptive immunocytes [1].

MALT is extensively distributed throughout our body, including GALT (gut-associated lymphoid tissue), NALT (nasopharynx- or nose-associated lymphoid tissue), BALT (bronchus-associated lymphoid tissue), and GENALT (genital-associated lymphoid tissue) [2]. In general, MALT’s anatomy consists of organized lymphoid follicles, which are predominantly formed by B-cell-dependent areas embedded in a network of follicular dendritic cells (FDC); CD4+ T lymphocytes and macrophages, and T-cell-dependent interfollicular areas predominantly containing CD4+ and CD8+ T lymphocytes [3]. The epithelium overlying MALT is follicle-associated epithelium (FAE) containing morphologically distinct cells named “microfold” or “membrane” (M), which are specialized in exogenous antigen sampling from the mucosal surfaces. Meanwhile, some intra- or subepithelial DCs can also capture antigens at the effector site (for example, nasal mucosa) and migrate into local/regional lymph nodes via draining lymphatics [4]. After the exogenous antigens are transported from the lumen to antigen-presenting cells, naïve T and B lymphocytes are effectively primed and migrate from the MALT to the peripheral blood, and subsequently, they are extravasated at the mucosal epithelium throughout the whole body [5]. Of note, mucosal tissues contribute approximately eighty percent of the total immunocytes in an adult person [6]. Vaccination is the most cost-effective strategy for preventing and controlling infectious diseases, but traditional systemic immunization cannot effectively elicit mucosal immunity. Given that the mucosal immune system has substantial roles in combating pathogenic microbes, the next generation of vaccines should focus on developing mucosa-targeting vaccines to effectively induce mucosal immune responses to confer superior protection.

## 2. Challenges and Solutions for the Development of Mucosa-Targeting Vaccines

Vaccine technology has advanced with huge achievements in the past few decades. COVID-19 vaccine development has been accomplished in less than one year since the COVID-19 outbreak. COVID-19 vaccines based on different technology platforms, including inactivated viruses, subunit proteins, viral-vectored vaccines, and mRNA-based vaccines, have been extensively administered for clinical use. Nevertheless, the current clinic-available vaccines are mostly used for inoculation via systemic vaccination (i.e., intramuscular or subcutaneous injections) [7]. While the conventional strategy of systemic vaccination effectively induces pathogen-specific antibody and/or cellular immune responses in the peripheral blood system, it is usually less effective in eliciting a strong protective immunity in the mucosal compartment. As mentioned above, the physical, chemical, and biological barrier of the mucosal surface can perform an effective defense function against pathogens’ invasion. However, these barriers also equally block the entry of traditional vaccines into the body. Thus, the challenges for the development of novel mucosa-targeting vaccines at least include (1) the insufficient uptake and breakthrough from the mucosal physical barrier into submucosa to activate mucosal immunity; (2) the fact that the robust degradation and digestion when passing through the gastrointestinal tract greatly limit the mucosal vaccines’ administration; (3) the fact that the conventional vaccines have poor adsorption and attachment onto the mucosal surfaces due to charge repulsion, causing a rapid clearance and thus lack of long-lasting antigen stimulation; and (4) the fact that the sampling of mucosal tissues is time-consuming, laborious, and high in technical requirements, so it is not easy to accurately measure the mucosal immune responses induced by mucosal vaccines.

To develop effective mucosal vaccines, numerous efforts and solutions are being pursued to address the above issues, including antigen administration through a needle-free mucosal route to induce mucosal immune responses, the development of mucosal adjuvants to enhance the effectiveness of mucosa-targeting vaccines, the construction of novel vectors as mucosa-targeting vaccines, nanoparticle-based formulations to reshape the vaccine–host cell recognition and then modulate the mucosal immunity, and antigen design to conjugate with mucosa-targeting ligands (Table 1). These strategies aim to break through the mucosal barrier to activate mucosal immunity. In this work, we summarize the latest findings on the development of mucosal vaccination strategies (Figure 2).

### 2.1. Antigen Administration through Needle-Free Mucosal Route to Induce Mucosal Immune Responses

Different from the conventional injection with syringes, mucosal vaccines are usually used through sprays, inhalation, oral administration, scratching, and patching through the respiratory tract, gastrointestinal tract, genital tract, and skin. The major advantage of mucosal inoculation is the better mimicking of natural infections through the mucosal surface, thereby potentially eliciting protective mucosal immunity. Importantly, mucosal inoculation can induce not only local mucosal immune responses at the inoculation site but also comprehensive mucosal immune responses at distal mucosal tissues through the common mucosal immune system [61]. Needle-free mucosal inoculation can reduce syringe use and medical waste production, which is a resource-saving and environmentally friendly strategy for the sustainable medical model. Moreover, it can be the Gospel and Savior for people who fear syringes. In addition, mucosal vaccines should be easier to make suitable for self-service inoculation, which would be very beneficial for the promotion of mass vaccination among the general population.

Intranasal inoculation has been increasingly developed to combat respiratory pathogens since it can induce a strong antigen-specific immune response in the nasal cavity/lung and then confer superior protection against respiratory infections [62,63,64,65,66,67,68,69,70]. For example, the intranasal administration of a live attenuated influenza vaccine (LAIV) has been clinically used in the USA since 2003. Furthermore, intranasal immunization can also elicit antigen-specific immune responses in the distal gastrointestinal tract and genital tracts. For example, the intranasal administration of the HIV-1 vaccine induced strong mucosal immune responses against HIV mucosal infections [8,9,10,11,12]. Additionally, inoculation via aerosol sprays or droplets is an attractive way to deliver COVID-19 vaccines [71,72].

Oral vaccination is also a common approach for mucosal vaccines. A well-documented oral vaccine is the orally attenuated poliovirus vaccine (OPV), and the OPV can effectively induce a strong mucosal immunity in the gastrointestinal tract, mammary gland, and salivary gland [13,14,15]. The orally administrated vaccines also include adenovirus types 4 and 7 vaccines (BL 125296/0), a rotavirus vaccine (RotarixTM, GlaxoSmithKline, Brentford, UK), a *Salmonella typhi* vaccine (Vivotif, Crucell Switzerland Ltd., Rehhagstrasse, Switzerland), and oral killed-cholera vaccines [16,17,18,19,20,21,22,73]. Additionally, oral vaccination with adenovirus-vectored vaccines could generate significant humoral and/or cellular immune responses against hepatitis B virus, rabies virus, measles virus, HIV, Ebola, and influenza virus [23,24,25,26,27,28,29].

Transcutaneous immunization (TCI) can penetrate the skin through skin scratches or microneedle patches, which can induce strong humoral and cellular responses in both systemic and mucosal compartments [74]. For example, smallpox had been eradicated from Earth by skin-scratch inoculation with the cowpox vaccine. Recently, microneedle patches have been extensively developed as a novel strategy to deliver different vaccines to induce mucosal immunity against the measles virus, malaria, and influenza [30,75].

In addition, sublingual (SL) immunization is gaining increased interest because of its optimal ability to induce immune responses in multiple mucosal and systemic tissues [31]. Eyedrop administration has also been demonstrated to effectively induce mucosal immune responses in both the ocular and nasal cavities [32,33]. Rectal immunization and vaginal immunization often evoke a strong local immune response in the rectum and genital tract and, thus, are suitable for preventing sexually transmitted diseases [34,35].

### 2.2. Development of Mucosal Adjuvants to Enhance the Effectiveness of Mucosa-Targeting Vaccines

An optimal adjuvant can improve the vaccine’s efficacy, modulate the type of immune response, prolong the protection time, and/or stabilize the vaccine’s formulation [76]. Various adjuvants based on different mechanisms are under extensive investigation. Among them, aluminum salt is the most well-known licensed adjuvant. It has been widely used in conventional inactivated vaccines or subunit vaccines to improve systemic immune responses, but it is less effective for inducing local IgA production and immunocyte homing into the mucosal compartment [77]. Therefore, it is of great interest to explore appropriate adjuvants for mucosal vaccine development.

Recently, the ongoing mucosal adjuvants include adenosine diphosphate (ADP) and bacterial enterotoxins—cholera toxin (CT) and Escherichia coli heat-labile toxin (LT) [36]. With the co-administration of pathogen antigens, these adjuvants induced antigen-specific IgA antibodies and long-lasting memory immune cells at the mucosal tissues, but the toxicity of CT and LT is a safety concern for their clinical use. The host recognition of harmful pathogens is accomplished by the pattern recognition receptors (PRRs) on the surfaces of host cells and the pathogen-associated molecular pattern (PAMP) on the surface of the pathogenic microbes. As a result, PAMP analogs (i.e., lipopolysaccharide, peptidoglycan, lipoprotein, teichoic acid, and bacterial DNA) and PRR agonists (i.e., various TLR ligands) could be promising adjuvants with which to enhance both systemic and mucosal immune responses [37,38]. For example, some TLR agonists such as CpG oligodeoxynucleotides (CpG-ODN), poly I:C, flagellin, R848, and Pam3CSK4, had been reported to effectively induce mucosal immunity as potential mucosal adjuvants [39,40]. In addition, some cytokines including interferons (IFNs), granulocyte macrophage-colony stimulating factor (GM-CSF), and interleukins (ILs) could also be used as mucosal adjuvants to increase IgG and IgA titers and/or local CTL activity [40,41].

### 2.3. Construction of Novel Vectors as Mucosa-Targeting Vaccines

The antigen-delivery system is another key prerequisite for developing an effective vaccine. Currently, various vaccines based on inactivated/protein subunits, recombinant viral vectors, bacterial vectors, DNA vectors, and the mRNA modality have been extensively studied. The various vectors usually elicit different profiles of immune responses and thus contribute to discrepant protection efficacy, and some vectors might be preferentially developed as mucosa-targeting vaccines because of their unique properties. For example, adenovirus type 5 (Ad5) is a common respiratory virus, and the recombinant Ad5-based vector has been extensively developed as vaccine candidates against SARS-CoV-2, influenza, Ebola, HIV-1, and other infectious diseases. Of note, mucosal vaccination (i.e., nasal) with Ad5-vectored vaccines could confer superior mucosal immunity and protection efficacy compared to systemic immunization. Similarly, the influenza virus is also a promising mucosal vector. Recombinant live attenuated influenza expressing an RSV G-protein domain induced a robust G-specific immune response in the lung and bronchoalveolar fluid and thus protected against RSV challenge in mice [48]. Our recent work also showed that intranasal inoculation with a replication-competent influenza vector carrying the HIV-1 P24 gene induced HIV-specific immune responses in the airway and vaginal tract in mice [49]. However, for these viral-vector-based vaccines, pre-existing anti-vector immune responses in the general population remain a challenge for their clinical application [78,79].

In addition, the intranasal or oral administration of a baculovirus-vectored human papillomavirus (HPV) vaccine conferred protection against vaginal HPV infection [50,51]. *Mycobacterium bovis* Bacillus Calmette–Guérin (BCG), the only licensed vaccine against TB, has been further developed as vaccine vectors against HIV-1 and SARS-CoV-2 [52,80,81,82].

### 2.4. Nanoparticle-Based Formulation to Reshape the Vaccine–Host Cell Recognition and Then Modulate the Mucosal Immunity

Interestingly, compared to soluble antigens, insoluble granular antigens at the mucosal surface can be taken in more efficiently through transcytosis or phagocytosis by M cells in the mucosal lymphoid tissue. Therefore, particulate formulations, such as virus-like particles (VLPs), bacterial ghosts, biodegradable nanoparticles, and immune-stimulating complexes, can benefit the efficacy of mucosal vaccines [53]. Such a mucosal delivery system can protect antigens from degradation, increase the attachment and absorption of antigens onto the mucosal surface, and prolong the residence time at local mucosal regions [42,43,44,45,46]. A recent study showed that the intranasal administration of a mixture of VLPs individually displaying H1, H3, H5, and H7 hemagglutinin (HA) epitopes significantly protected mice against hetero-variant or hetero-subtypic influenza challenge [54]. Moreover, particles encapsulated with mucoadhesive and biodegradable polymer particles, such as chitosan, polyethyleneimine (PEI), poly lactic-co-glycolic acid (PLGA), glycolides, epoxy polymers, hydrogels, and paraffin, have also been used in the development of mucosal vaccines for HIV-1, TB, and malaria [55]. Lipid-based particles such as liposomes, archaeosomes, niosomes, virosomes, ISCOMs, microbubbles, and emulsions have been tested in animal models for mucosal vaccines [56,57]. For example, the intranasal administration of an anionic mRNA encoding the envelope glycoprotein gp120 of HIV-1 with a PEI–cyclodextrin polymer prolonged the nasal residence time and increased the uptake of the mRNA vaccine by nasal epithelial cells, and thus, the gp120-specific immune responses in serum and mucosal samples were significantly boosted [83]. In addition, our recent study showed that a chitooligosaccharide shell enabled the adenovirus-vectored HIV vaccine to enhance mucoadhesion to nasal tissues and elicited strong IgA production and T-cell immunity at local and remote MALT in mice [47].

### 2.5. Antigen Design to Conjugate with Mucosa-Targeting Ligands

As mentioned above, M cells and DC cells in MALT play a key role in antigen uptake and antigen presentation, and thus, it is reasonable to design antigens to target these cells to enhance the mucosal immune responses. M-cell-targeted ligands include ulex europaeus agglutinin 1 (UEA-1), FimH, and membrane protein H (OmpH), which can bind to the α-L-fucose residues, transcytotic receptor glycoprotein 2 (GP2), and C5a receptor (C5aR) on the M cells in MALT. DC-targeted ligands include the TLR family, Clec9A, Clec12A, DEC205, MHCII, CD11c, FcγR and PD-L1 [47,84,85].

In general, the local mucosal immunity induced at the vaccination site is stronger than that at the distal mucosal site. One explanation might be the lymphocytes homing to the endothelial receptors in the corresponding mucosal tissue, which is induced by the imprinting of DC cells through the upregulation of the expression of tissue-specific adhesion molecules and chemokine receptors on lymphocytes [86]. For example, vaccine antigens in the intestinal mucosa are often taken up by intestinal mucosa-specific DC cells, which act as imprinting cells and up-regulate the expression of lymphocyte surface homing receptor α4β7-integrin and CCR9 molecules. A4β7-integrin and CCR9 can strongly interact with MADCAM1 on the venules in the intestine and CCL25 in the epithelial cells of the intestine [58]. Skin-derived DCs can imprint T cells to express P- and E-selectin ligands and CCR10, which would lead these T cells to preferentially home into the skin via P- and E-selectins and CCL27, respectively [87]. Moreover, IgA-secreting B cells in MALT can express CCR10, the receptor for CCL28, which is secreted by the epithelial cells throughout the intestines, salivary glands, tonsils, respiratory tract, and mammary glands [36]. Therefore, antigen design to conjugate with these molecules to regulate the DC imprinting effect on lymphocytes would effectively induce immune responses in certain mucosal sites. For example, RALDH2, as a molecular adjuvant, can regulate the homing of lymphocytes in intestinal mucosa to induce an effective intestinal mucosal immune response [59,60].

## 3. Conclusions and Perspective

As the most cost-effective strategy against infectious diseases, vaccination technology has substantially improved our public health. Currently, the clinically available vaccines are mostly aimed at inducing systemic immune responses, and only a very few licensed vaccines are designed to elicit local mucosal immunity instead. Of note, these few mucosal vaccines have already made incredible achievements in fighting infectious diseases. For instance, the highly lethal smallpox caused by variola virus had been eradicated by skin-scratch inoculation with the cowpox vaccine in the late 1970s; another terrible infectious disease—poliomyelitis, caused by poliovirus—might be eradicated with both an orally attenuated poliovirus vaccine (OPV) and the injection of an inactivated poliovirus vaccine (IPV) in the near future [88]. These achievements highlight the unique importance of further developing mucosal vaccine technology to combat the frequent outbreaks of infectious diseases.

The development of mucosa-targeting vaccines has been greatly limited due to the physical, chemical, and biological barriers of MALTs. The difficulties of mucosal tissues’ sampling and lack of surrogate biomarkers with which to assess mucosal immune responses also restrict the development of mucosal vaccines. To overcome these challenges, various strategies to improve the efficacy of mucosal vaccines have been rapidly developing in recent years, though their effectiveness should be further evaluated in clinical studies. Among them, intranasal vaccination is extensively thought of as a promising approach to eliciting mucosal immunity against respiratory pathogens, such as influenza and SARS-CoV-2.

Vaccinology is a well-known multidisciplinary science involving immunology, virology, clinical medicine, bioinformatics, chemistry, biomaterials, and nanoscience. These advanced interdisciplinary techniques are promoting the continuous innovation of vaccine technology, such as reverse vaccinology, nanovaccinology, structure-based vaccine design, and mRNA-based vaccines. Hopefully, these technology innovations will also greatly accelerate mucosal vaccine development. Altogether, it is of great significance to develop novel mucosa-targeting vaccines as the next generation of vaccine technology against emerging infectious diseases.

## Figures and Tables

**Figure 1 viruses-14-00520-f001:**
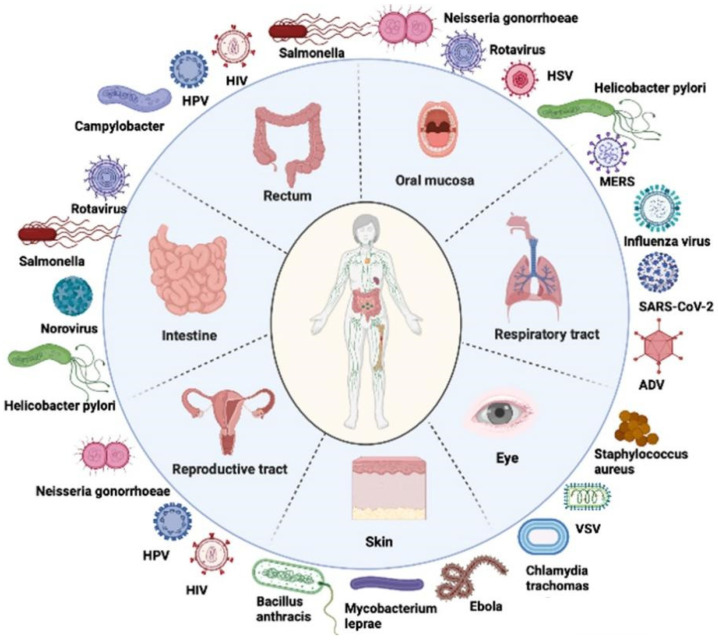
Numerous pathogenic microbes can invade the human body through different mucosal routes.

**Figure 2 viruses-14-00520-f002:**
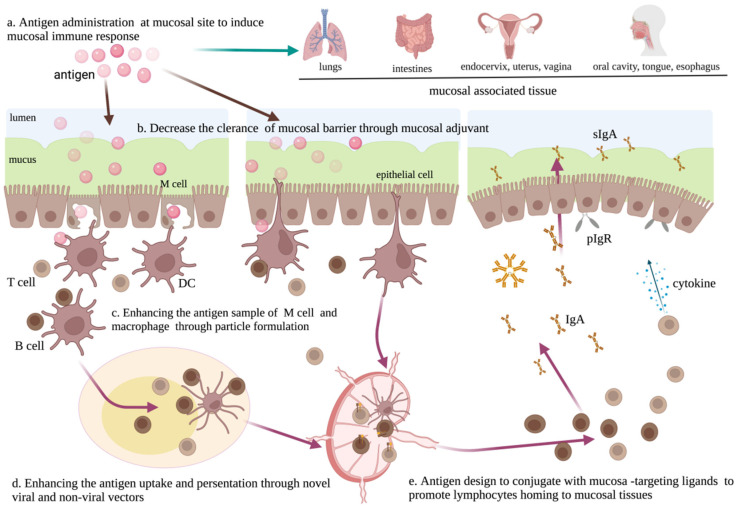
Challenges and solutions for the development of mucosa-targeting vaccines. (**a**) Antigen administration through mucosal route (such as aerosol inhalation, rectum, vagina, and oral administration) can effectively induce mucosal responses through mimicking the natural infection of pathogens. (**b**) Antigen administration along with mucosal adjuvants can enhance the mucosal immune response through avoiding the adverse effect of the mucosal barrier. (**c**) Encapsulating the vaccine into a nanoparticle formulation can enhance mucosal responses because M cells and DC cells preferentially sample particle antigens from the lumen. (**d**) The use of novel vaccine-delivery vectors, including viral and non-viral vectors, can effectively promote antigen uptake and presentation, and thus enhance mucosal responses. (**e**) Antigens conjugated with mucosa-targeting ligands can induce mucosal responses through homing to the mucosal tissues.

**Table 1 viruses-14-00520-t001:** Strategies with which to develop mucosa-targeting vaccines.

Strategies	Classification	Application	Advantages	Disadvantages	References
Vaccine administration through needle-free mucosal route	Spray, inhalation, oral administration, scratching, and patching through the respiratory tract, genital tract, skin, etc.	Adenovirus 4 and 7 (BL 125296/0), Salmonella typhi (Vivotif), smallpox, influenza viruses (LAIV), rotavirus (RotarixTM), cholera, respiratory syncytial virus, tuberculosis (RSV), pertussis, SARS-CoV, MERS-CoV and SARS-CoV-2, HIV, Ebola, Zika virus, etc.	Better mimic natural infections through the mucosal surface and induce not only local mucosal immune responses at the inoculation site but also comprehensive mucosal immune responses at distal mucosal tissues. Reduce syringe use and medical waste production. Relieve needle injection pain.	More easily blocked and degraded by the harsh mucosal barrier; weak mucosal immune responses.	[8,9,10,11,12,13,14,15,16,17,18,19,20,21,22,23,24,25,26,27,28,29,30,31,32,33,34,35]
Vaccine administration with mucosal adjuvants	Bacterial adjuvants such as CT, LT; TLR agonists including CpG ODN, Poly I:C, flagellin, R848, and Pam3CSK4; cytokine adjuvants such as IL1, GM-CSF, and IFNs; mucoadhesive polymers including shellac, cellulose acetate phthalate, cellulose acetate trimellitate, Eudragit, and polymers; polycarbophils, etc.	HPV, HBV, etc.	Enable antigens to evade clearance and trap mucosal barriers to evoke a stronger mucosal immune response. Protecting antigens from degradation, increasing concentration of antigen in the vicinity of mucosal tissue for better absorption, extending their residence time in the body and/or releasing them at a specific mucosal site.	Safety problems brought by adjuvant components may restrict the development of vaccines; the stability of the antigen may be influenced, and the preparation may become complicated and expensive.	[36,37,38,39,40,41,42,43,44,45,46,47]
Development of novel mucosal vectors	Live-attenuated vaccine, recombinant replicating adenovirus vector, baculovirus vector, BCG vector, influenza vector, etc.	Influenza, HPV HIV, SARS-CoV-2, RSV, and HIV	Easily induce mucosal immune responses without adjuvant assistance.	Stability and safety need to be improved, and new vectors need clinical validation	[48,49,50,51,52]
Nanoparticle-based formulation to reshape the mucosal immunity	VLPs, bacterial ghosts, and immune-stimulating complexes; biodegradable micro-/nanoparticles including PLGA, glycolides, epoxy polymers, hydrogels, paraffin, etc.; lipid-based particles including liposomes, archaeosomes, niosomes, virosomes, ISCOMs, microbubbles, emulsions, etc.	HIV, TB, and malaria	Antigens delivered in particles are better recognized by the innate immune system, and better captured by M cells and DCs; thus, a stronger mucosal immune response can be induced.	Particle formulations require the assistance of polymers or liposomes and, thus, are subject to the development and influence of chemical materials.	[53,54,55,56,57]
Antigen design to conjugate with mucosa-targeting ligands	M-cell-targeted ligands including UEA-1, FimH, and OmpH; DC-targeted ligands including specific antibodies and agents directed against DC receptors such as TLR family, Clec9A, Clec12A, DEC205, MHCII, CD11c, FcγR, etc.; mucosal-epithelial-cell-targeted ligands including transferrin, IgG Fc fragment, etc.;	HIV, RSV, etc.	Antigens coupled with ligands targeting M cells and DC cells are better captured by M cells and DCs; thus, a stronger mucosal immune response can be induced. It can directly target lymphocytes to the target mucosal site, thus precisely inducing an effective mucosal response at the target mucosal site.	Antigen design is complex, is still immature and needs to be validated in clinical trials.	[36,58,59,60]
	lymphocyte-migration-targeting molecules including α4β7-integrin, CCR9, CCL25, CCR10, CCR4, CCL20, CXCL9, CCL28, RALDH2, etc.				

## Data Availability

Not applicable.

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
