# Peer review of "Strategies to Develop a Mucosa-Targeting Vaccine against Emerging Infectious Diseases"

_viruses, 2022, doi:10.3390/v14030520_

Round 1
Reviewer 1 Report
In this review/perspective article the authors provide a comprehensive description of mucosa-targeting vaccines. A general comment on the manuscript would be for the authors to provide a more detailed description of how to tackle the challenges related to the development of these vaccines in the text (Lines 80-87, Figure 2).
Other minor comments on the manuscript are:
Line 11: please replace ‘penetrate’ with ‘infect’.
Please avoid using ‘etc’ and ‘so on’ in the text (e.g. Lines 18, 22, 31, 39, 97, 122, 148, 262, 216).
Lines 52-55: please rephrase this sentence to generalize and provide an appropriate literature reference.
Lines 55-56: please provide a literature reference.
Lines 67-69: I would suggest to the authors to avoid using emotionally charged language (‘One miracle is...’).
Line 78: please replace ‘Embarrassingly’ with another connective word that indicates contrast.
Lines 99-102: please provide a literature reference.
Line 238: clinically available
Lines 243-246: to what data do the authors base these claims? Please add the appropriate references.
Author Response
In this review/perspective article the authors provide a comprehensive description of mucosa-targeting vaccines. A general comment on the manuscript would be for the authors to provide a more detailed description of how to tackle the challenges related to the development of these vaccines in the text (Lines 80-87, Figure 2).
Response: Thank you for your kind comments and suggestions, and we have modified these descriptions accordingly in the revised manuscript (line113-120, page 3; line123-132, page 3).
Other minor comments on the manuscript are:
Line 11: please replace ‘penetrate’ with ‘infect’.
Response: Thank you for your correction, and we have replaced “penetrate” with “infect” in revised manuscript (line11, page 1).
Please avoid using ‘etc’ and ‘so on’ in the text (e.g. Lines 18, 22, 31, 39, 97, 122, 148, 262, 216).
Response: Thank you for your kind reminders, and we have deleted them accordingly in revised manuscript.
Lines 52-55: please rephrase this sentence to generalize and provide an appropriate literature reference.
Response: Thank you for your kind suggestions, we have rephrased this sentence and modified the description accordingly in revised manuscript (line60-63, page 2).
Lines 55-56: please provide a literature reference.
Response: Thank you for your reminder, and we have added the related references in revised manuscript (line63-65, page 2).
Lines 67-69: I would suggest to the authors to avoid using emotionally charged language (‘One miracle is...’).
Response: Thank you for your kind mention, and we have rephrased this sentence in revised manuscript (line 75-77, page 2).
Line 78: please replace ‘Embarrassingly’ with another connective word that indicates contrast.
Response: Thank you for your kind mention, and we have replaced “Embarrassingly” with “However” in revised manuscript (line 103, page 3).
Lines 99-102: please provide a literature reference.
Response: Thank you for your mention, and we have added the related reference in revised manuscript (line142-145, page 4).
Line 238: clinically available
Response: We have replaced “clinical-available” with “clinically available” in revised manuscript (line 290, page 6).
Lines 243-246: to what data do the authors base these claims? Please add the appropriate references.
Response: Thank you for your kind mention, we have added the related reference in revised manuscript (line297-300, page 7).
Reviewer 2 Report
Strategies to develop the mucosa-targeting vaccine against emerging infectious diseases
Dear author and editor:
This review perceptive article talked about the importance of vaccines that target mucosa and showed different strategies used in developing mucosa-targeting vaccines.
The article could be published after a minor revision
I have some comments on it:
- The legend of the figure 2 doesn’t explain well the figure. It needs more explanation.
- According to the literature, which is the best method of antigen delivery which can elicit mucosal immune response.
- The author talked about viral and non viral vector in mucosa- targeting vaccine. Is there immune response against the viral vector could affect the vaccines activity and its biodegradation ?
- Do you think that using monoclonal antibody or nanobodies could be alternative strategies to block mucosal entry?
Thank you very much, all the best

Author Response
This review perceptive article talked about the importance of vaccines that target mucosa and showed different strategies used in developing mucosa-targeting vaccines. The article could be published after a minor revision.
Response: Thank you for your positive comments to our work.
I have some comments on it:
- The legend of the figure 2 doesn’t explain well the figure. It needs more explanation.
Response: Thank you for your kind suggestion, and we have added the detailed description to explain it more clearly in the revised manuscript (line 123-132, page 3).
- According to the literature, which is the best method of antigen delivery which can elicit mucosal immune response.
Response: Thank you for your question. Recently, various strategies have been rapidly developing to improve the efficacy of mucosal vaccines, including novel needle-free mucosal-targeting route, development of mucosal-targeting vectors, administration of mucosal adjuvants, encapsulating vaccine into nanoparticle formulation, antigen design to conjugate with mucosa-targeting ligands. It is hard to predict the most appropriate delivery technology to make an effective mucosal vaccine, as many strategies are still developing and their effectiveness should be further evaluated in clinical studies. Nevertheless, according to reported literature, intranasal vaccination is extensively thought as a promising approach to elicit mucosal immunity against respiratory pathogens, such as influenza, SARS-CoV-2. We have mentioned this description in the revised manuscript. “To overcome these challenges, various strategies have been rapidly developing to improve the efficacy of mucosal vaccines in recent years, though their effectiveness should be further evaluated in clinical studies. Among them, intranasal vaccination is extensively thought as a promising approach to elicit mucosal immunity against respiratory pathogens, such as influenza, SARS-CoV-2.” (line 305-309, page 7)
- The author talked about viral and non viral vector in mucosa- targeting vaccine. Is there immune response against the viral vector could affect the vaccines activity and its biodegradation?
Response: Thanks for your kind mention. We have mentioned this point in our revised manuscript. “However, as for these vector-based vaccines, pre-existing anti-vector immune responses in general population remain a challenge for its clinical application”. (line 228-229, page 5)
- Do you think that using monoclonal antibody or nanobodies could be alternative strategies to block mucosal entry?
Response: Thank you for your kind comment. As you mentioned, monoclonal antibodies or nanobodies can also be used as an antiviral tool to block mucosal entry. However, antibodies are usually very expensive, and also rapidly metabolized in a short time. Therefore, it is not suitable to use antibodies for a long-term prophylaxis tool. In general, a convenient and long-lasting protection should depend on the effective mucosal vaccines. Thank you.